# Cholinergic-estrogen interaction is associated with the effect of education on attenuating cognitive sex differences in a Thai healthy population

Chen Chen[1,2], Bupachad Khanthiyong[3], Sawanya Charoenlappanit[4], Sittiruk Roytrakul[4], Gavin P. Reynolds[5], Samur Thanoi [6] *, Sutisa Nudmamud-Thanoi [2,7] *

1 Medical Science Graduate Program, Faculty of Medical Science, Naresuan University, Phitsanulok, Thailand, 2 Centre of Excellence in Medical Biotechnology, Faculty of Medical Science, Naresuan University, Phitsanulok, Thailand, 3 Faculty of Medicine, Bangkokthonburi University, Bangkok, Thailand, 4 Functional Proteomics Technology Laboratory, National Centre for Genetic Engineering and Biotechnology, National Science and Technology Development Agency, Pathum Thani, Thailand, 5 Biomolecular Sciences Research Centre, Sheffield Hallam University, Sheffield, United Kingdom, 6 School of Medical Sciences, University of Phayao, Mae Ka, Phayao, Thailand, 7 Department of Anatomy, Faculty of Medical Science, Naresuan University, Phitsanulok, Thailand

* sutisat@nu.ac.th (SNT); samur.t@up.ac.th (ST)

**Data Availability Statement:** All relevant data are within the paper and its Supporting information files.

## Abstract

The development of human brain is shaped by both genetic and environmental factors. Sex differences in cognitive function have been found in humans as a result of sexual dimorphism in neural information transmission. Numerous studies have reported the positive effects of education on cognitive functions. However, little work has investigated the effect of education on attenuating cognitive sex differences and the neural mechanisms behind it based on healthy population. In this study, the Wisconsin Card Sorting Test (WCST) was employed to examine sex differences in cognitive function in 135 Thai healthy subjects, and label-free quantitative proteomic method and bioinformatic analysis were used to study sex-specific neurotransmission-related protein expression profiles. The results showed sex differences in two WCST sub-scores: percentage of Total corrects and Total errors in the primary education group (Bayes factor>100) with males performed better, while such differences eliminated in secondary and tertiary education levels. Moreover, 11 differentially expressed proteins (DEPs) between men and women (FDR<0.1) were presented in both education groups, with majority of them upregulated in females. Half of those DEPs interacted directly with nAChR3, whereas the other DEPs were indirectly connected to the cholinergic pathways through interaction with estrogen. These findings provided a preliminary indication that a cholinergic-estrogen interaction relates to, and might underpin, the effect of education on attenuating cognitive sex differences in a Thai healthy population.

**Funding:** Chen Chen received financial support from the Naresuan Competitive Grants for International Students (NCG) for studying doctoral degree and Sutisa Nudmamud-Thanoi received partial support from the Reinventing University Program 2023, the Ministry of Higher Education, Science, Research and Innovation (MHESI), Thailand (R2566A044), and the NCG grant has no grant number provided. The funders had no role in study design, data collection and analysis, the decision to publish, or the preparation of the manuscript.

**Competing interests:** The authors have declared that no competing interest exist.

## Introduction

Human brain development is influenced by both genetic and environmental factors. The X and Y chromosomes define not just a person's sex, but also the morphological and functional distinctions between male and female brains. For example, the male brain tends to be bigger with a higher proportion of white matter [1], whereas the female brain generally has a larger corpus callosum [2]. According to Ingalhalikar et al. [3], the male brain is thought to be more intrahemispherically connected, whereas the female brain appears optimized for interhemispheric connectivity. There is growing acknowledgment that the human brain is sensitive to environmental circumstances throughout development, such as socioeconomic status, education, nutrition, and parental care [4]. Evidence suggests that children from higher-income families have greater volumes of grey and white matter in the inferior frontal gyrus and bigger hippocampus sizes than their lower socioeconomic status peers [5, 6]. Furthermore, education not only enhances human brain development but also attenuates cognitive deficits that occur with aging [7–9]. Schooling increases cortical grey matter volumes (GMV) in several brain areas [10], and even a few years of education has been demonstrated to contribute to brain cognitive reserve [11].

Under the joint actions of genetic and environmental factors, male and female brains may differ in terms of cognitive strategies and/or cognitive styles [12]. These include different learning style preferences [13], as well as sex differences in decision-making [14]. Previous research found that in correct trials of the Wisconsin Card Sorting Test (WCST), males had shorter response time, which is a measure of efficiency of the decision-making process [15], while females made fewer WCST errors [16]. However, another study conducted by Cinciute and colleagues [17] found no significant difference in the duration of WCST task performance regarding sex.

There has been minimal research into the effect of education on attenuating cognitive sex differences. Bloomberg et al. [18] discovered that males in UK had higher fluency scores only in the low education group (less than high school diploma). There was evidence of higher fluency scores in females in the high education group (high school diploma and higher), especially in those born later. This finding implied that schooling, as well as secular changes in education level across birth cohorts, had a role in determining cognitive performance in females. However, it is unknown whether this result extends to other ethnicities or cognitive domains.

Cognitive performance is the reflection of information transmitted along with neural circuits of brain areas [19, 20], and learning and obtained experiences have been documented to alter the morphology of synapses as well as promote their plasticity [21]. The study of how education affects neurotransmission in male and female brains is still in its infancy and has to be expanded. Therefore, the purpose of this study is to investigate the effects of education on cognitive sex differences in a Thai healthy population, as well as to investigate the neural mechanisms behind such effects.

## Materials and methods

### Study subjects

As described in a previous study from our research team [22], one hundred and thirty-five healthy volunteers between the ages of 22 and 70 were recruited. Subjects were divided into two categories based on their education level: those who had acquired no more than primary education, and those who had received secondary education, for some, tertiary education. Subjects with abnormal mental health evaluated by the Thai Mental Health Indicator (THMI-55)

were excluded from the current study. The Mini-Mental State Examination (MMSE) was also used to screen out participants with dementia. To limit the likelihood of confounding by population stratification, all individuals were of Thai ethnicity [23]. Written consent forms were obtained from the participants involved. The experimental protocols were approved by the Human Ethics Committee of Naresuan University (Naresuan University Institutional Review Board), COA No. 0262/2022.

## Cognitive assessment

The Wisconsin Card Sorting Test (WCST) has long been used in clinical and research settings to assess the frontal cortex (FC) function [24]. The recent neuroimaging study revealed that, in addition to FC, a widespread network of prefrontal, frontal, temporal, frontal-temporal, and parieto-occipital regions are activated during WCST [25], and multiple domains of cognitive function including but not limited to attention, cognitive set-shifting, planning or organizing, problem-solving and decision-making are activated during the process of WCST [26].

In this study, all subjects were tested using a computer-based WCST (Inquisit 3.0.6.0), and the raw score was analyzed to reflect different domains of cognitive function, as shown below [27]:

- The percentage of total corrects (%Corrects): the entire number of correct response cards multiplied by 100 and divided by the total number of cards, reflecting initial conceptualization and attention.

- The percentage of total errors (%Errors): the total number of incorrect response cards multiplied by 100 and divided by the entire number of cards, reflecting nonspecific cognitive impairment.

- The number of categories completed (Category completed): determined by applying the score range from 1 to 6, reflecting cognitive set-shifting.

- The perseverative errors (PE): the score was used to assess the inability to correct a response due to ignorance of relevant stimuli, reflecting cognitive inflexibility.

- Trails to complete the first category (1st Category): the score ranges from 0 to 128, which is the number required to complete the initial category of the task, reflecting initial conceptualization.

## Blood sample collection

A 3ml of cubital vein blood was collected from each enrolled subject immediately after they completed the WCST test. The blood was centrifuged at 3,000 rpm for 5 minutes. The serum was then transferred into a 1.5ml microcentrifuge tube and stored at -80˚C in a refrigerator for future use, all samples were coded to ensure anonymity.

## Label-free quantitative proteomics analysis

The label-free quantitative proteomics analysis was used to compare serum protein expression profiles between males and females. The analytic processes were performed by the Functional Proteomics Technology Laboratory, National Centre for Genetic Engineering and Biotechnology, Pathum Thani, Thailand, which include protein digestion, Liquid Chromatography with tandem mass spectrometry (LC-MS/MS) analysis, protein identification, and protein quantitation.

In brief, the protein concentration of all serum samples was determined using the Lowry assay with BSA as a standard protein [28]. Five micrograms of protein samples were digested with sequencing grade porcine trypsin (1:20 ratio) for 16 hours at 37 ˚C. The tryptic peptides were dried in a speed vacuum concentrator and resuspended in 0.1% formic before further analysis.

The prepared tryptic peptide sample of each subject was injected individually into an Ultimate3000 Nano/Capillary LC System (Thermo Scientific, UK) coupled to a Hybrid quadrupole Q-Tof impact II™ (Bruker Daltonics) equipped with a Nano-captive spray ion source. Mass spectra (MS) and MS/MS spectra were obtained in the positive-ion mode at 2 Hz over the range (m/z) 150–2200. To minimize the effect of experimental variation, three independent MS/MS runs were performed for each sample.

MaxQuant 1.6.6.0 was used to quantify and identify the proteins in each sample with the Andromeda search engine to correlate MS/MS spectra to the Uniprot *Homo sapiens* database [29]. The proteins were identified using a 10% protein false discovery rate (FDR), carbamido-methylation of cysteine as fixed modification, and the oxidation of methionine and acetylation of the protein N-terminus as variable modifications. Only proteins with at least two peptides, and at least one unique peptide, were considered as being identified and used for further data processing.

### Bioinformatic analysis

Before any analysis, data cleansing and preprocessing were performed by using Perseus ver. 1.6.15.0 [30]. A list containing all proteins identified previously was submitted to jvenn (web application) to explore those proteins shared by males and females in both primary and secondary and tertiary education groups [31]. Differentially expressed proteins (DEPs) between males and females in each education group were detected independently using the Linear Model for Microarray Data (LIMMA) approach within R-programming ver. 4.1.2 [32], with the FDR set at 10%. Multi-Experiment Viewer (MeV, ver.4.9.0) software was employed to illustrate the expressions of shared DEPs in men and women from both education groups [33]. The protein-protein interaction (PPI) network, as well as the relationship between DEPs and cognitive function, was investigated using Pathway Studio ver. 12.5 [34, 35].

### Statistical analysis

Using R-programming ver. 4.1.2, the multi-factorial analysis of variance (ANOVA) was performed to study the main effects of sex, age and education, as well as their interactions, on WCST scores. A General Linear Model (GLM) approach paired with Bayesian statistics was applied to analyze differences in WCST scores between men and women at each education level [36], as well as comparisons of males and females across the two education levels, with age as a covariate. Bayes factor (BF) was offered to illustrate the likelihood of supporting the alternative hypothesis [37, 38], and the sensitivity analysis was also employed to test the robustness of the results [39]. BF≥10 or P≤0.05 was considered significant in this study.

## Results

### Demographic characteristics of the study population

The Subjects were 70 males and 65 females, with a mean age of 57.75±10.35 years (range, 22–70 years). Table 1 describes their demographic data, including sex, age, and educational level. The study found that there were significant age differences between males and females in the secondary and tertiary education groups, but not in the primary education group. However,

**Table 1. Demographic data of subjects.**

|  | Male | Female | BF | 95% CI |
|---|---|---|---|---|
| **Primary education** |  |  |  |  |
| No. of subjects | 49 (70%) | 42 (64.6%) |  |  |
| Age | 61.4±6.02 | 62.5±6.81 | 1.56 | [-3.78, 1.66] |
| **Secondary and Tertiary education** |  |  |  |  |
| No. of subjects | 21 (30%) | 23 (35.4%) |  |  |
| Age | 51.1±13.2 | 47.3±9.75 | 5.73 | [-3.18, 10.88] |

Data was presented as mean±SD by General Linear Model.

BF = Bayes Factor, 95% CI = 95% Credible interval of difference male against female.

age difference between the two education groups was discovered with decisive evidence (BF>100, 95%CI = [9.65, 16.0]).

## The main effects of sex, age, and education, as well as their interactions, on WCST scores

The results of multi-factorial ANOVA revealed that sex, age and education level had significant main effects on two of the five WCST-sub scores %Corrects and %Errors (see Table 2). Aside from that, there were significant interaction effects of sex by age (P<0.01) on scores %Corrects, %Errors and Category Completed, as well as significant age by education interaction effect on 1st Category (P<0.05).

## Effects of education on sex differences in cognitive performance

The Jeffreys-Zellner-Siow (JZS) priors [40] is the default prior used in the general linear model, with a default r scale of 0.353. The data suggested that education have potential compensatory effects on three out of the five WCST sub-scores: %Corrects, %Errors, and 1st Category. In the primary education group, male scored better in %Corrects (BF>100, 95%CI = [2.74, 12.25], Cohen's d = 0.68) and had lower %Errors (BF>100, 95%CI = [-12.2, -2.71], Cohen's d = 0.67) with close to large effect size [41]. While sex differences in those two scores reversed in the higher education group, however, there was only weak evidence to support the alternative hypotheses (both BF<2) (see Fig 1 and full data set in S1 Table). Regarding the score 1st Category, females were found to have fewer trials to complete the first category in the WCST test compared to males in both education groups, but there was weak evidence to reject the null hypothesis in the lower education group (BF = 1.02). Such sex difference was decisive in the second and tertiary education group (BF>100, 95%CI = [-6.65, 15.8]), although the effect size was small (Cohen's d = 0.25). The results remain robust after sensitivity analyses.

**Table 2. Main effects of sex, age and education on WCST sub-scores.**

|  | %Corrects | %Errors | 1st Category | Category Completed | PE |
|---|---|---|---|---|---|
| Sex | P = 0.05* | P = 0.05* | P = 0.32 | P = 0.73 | P = 0.08 |
| Age | P = 0.01* | P = 0.01* | P = 0.50 | P = 0.26 | P = 0.68 |
| Education | P = 0.03* | P = 0.02* | P = 0.26 | P = 0.58 | P = 0.28 |

*P≤0.05 by multi-factorial ANOVA

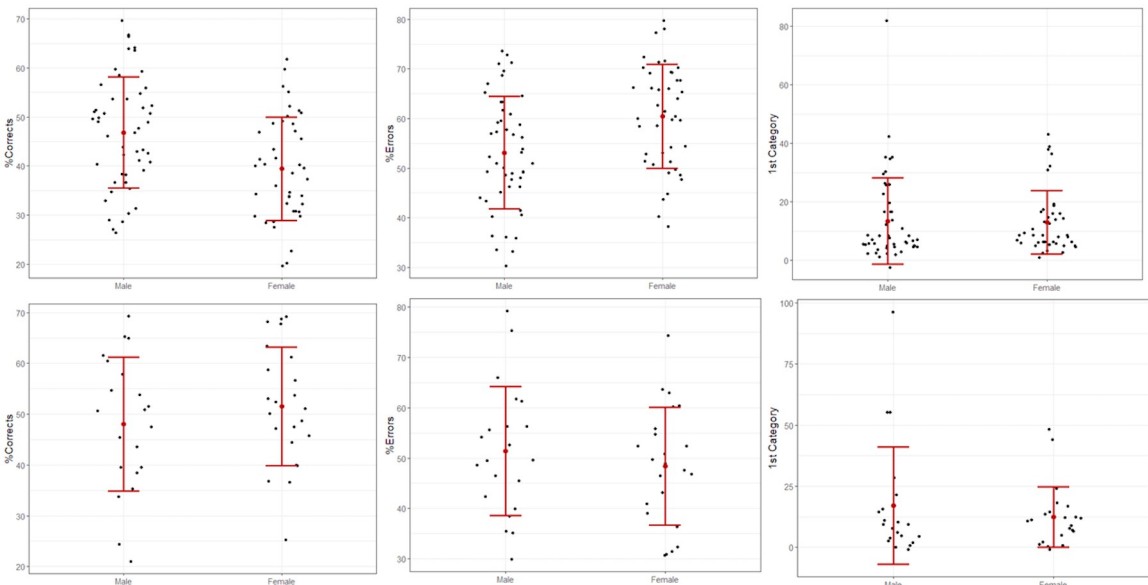

**Fig 1. Effects of education on sex differences in WCST sub-score %Corrects, %Errors, and 1<sup>st</sup> Category.** The upper graphs show sex differences in those scores in the primary education group, the lower graphs depict the sex differences in the same scores in the secondary and tertiary education groups.

Furthermore, there was insufficient evidence to accept the sex differences in score PE and Category Completed in both education groups (all BF<6).

In addition, females with secondary and tertiary education performed significantly better in scores %Corrects and %Errors (BF>100) than those in the lower education group, and moderate evidence for such difference in the score 1<sup>st</sup> Category (BF = 4.55). While limited evidence indicated differences in scores of %Corrects (BF = 4.79), %Errors (BF = 4.43), and 1<sup>st</sup> Category (BF = 3.48) in males across the two education levels.

## Identification and relative quantification of differentially expressed proteins between men and women

After applying the criteria outlined in section 2.4, a total of 886 proteins were identified using a label-free quantitative proteomics approach, with 808 of them shared by both sexes (see Fig 2). Of those 808 proteins, we found that 11 differentially expressed proteins (DEPs) between men and women (FDR<0.1) were present in both education groups (see Fig 3 and full data set in S2 Table). Except for Hematopoietic progenitor cell antigen CD34 (CD34), all of the 11 DEPs were upregulated in females.

## Protein-protein interactions of DEPs and their relationship with cognitive function

As shown in Fig 4, out of the 11 DEPs, LRP4, TTR, TRIM38, and PP2B interacted directly with nAChR3, a subunit of nicotinic acetylcholine receptor that positively regulates cognition. Of the others, CD34, SERPING1, HSD17B1, KNG1, and TTR are indirectly connected to the cholinergic pathways through interaction with estrogen. This suggested that the cholinergic-estrogen interaction may have an influence on cognitive processes. STRCP1 and RAB4A, on the other hand, showed no interaction with other DEPs or association with cognitive function.

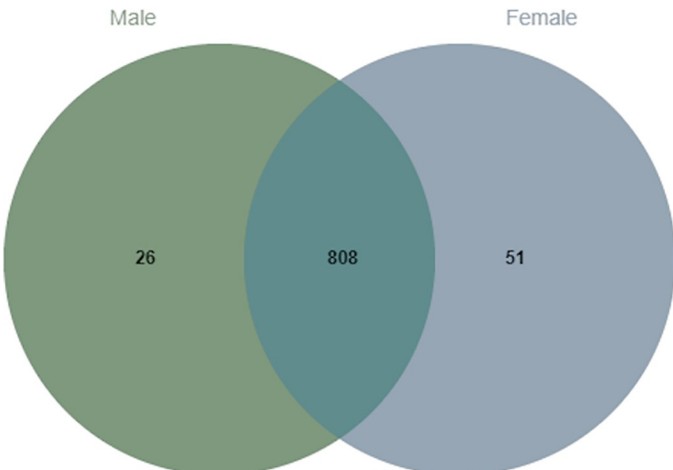

**Fig 2. Venn diagram showing 808 proteins shared by both males and females.**

## Discussion

In this study, we observed sex differences in WCST sub-scores %Corrects and %Errors in the primary education group, with males outperforming females, which is consistent with earlier research [42], but these differences were not supported by sufficient evidence in the higher education group. The WCST is a test of cognitive flexibility, that is the ability to adjust behavioral response mode in the face of changing conditions [25, 43]. Its sub-scores %Corrects and %Errors show specific domains of cognitive function, which correspond to the activity of certain brain regions. For example, %Corrects reflects conceptualization and attention, and a previous study showed that multiple areas of the prefrontal cortex (PFC) may participate in information processing during attentional shift in the WCST [44, 45]. While %Errors reflecting non-specific cognitive impairment and frontal and temporal lobes lesions have been linked to a large number of WCST total errors [46, 47].

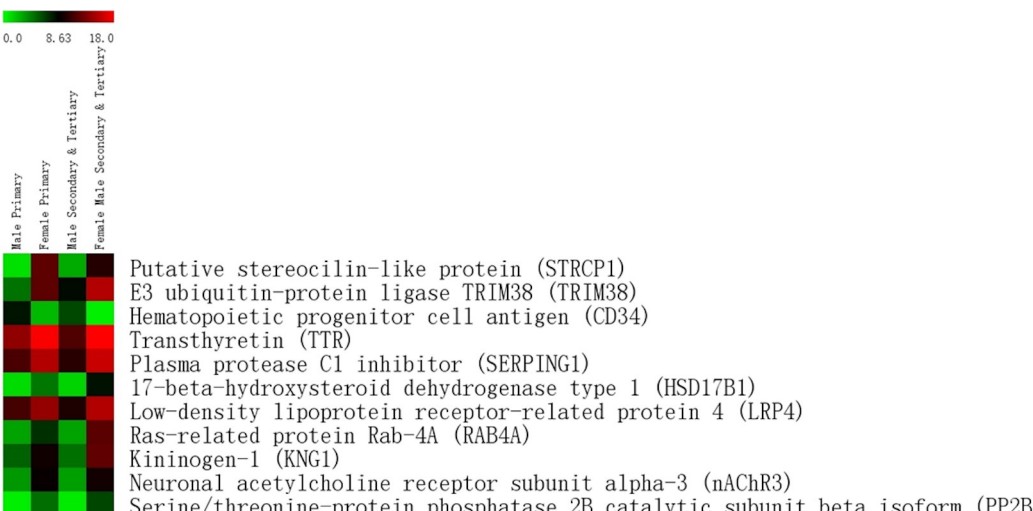

**Fig 3. Expression heatmap of 11 proteins exhibited in both education groups.**

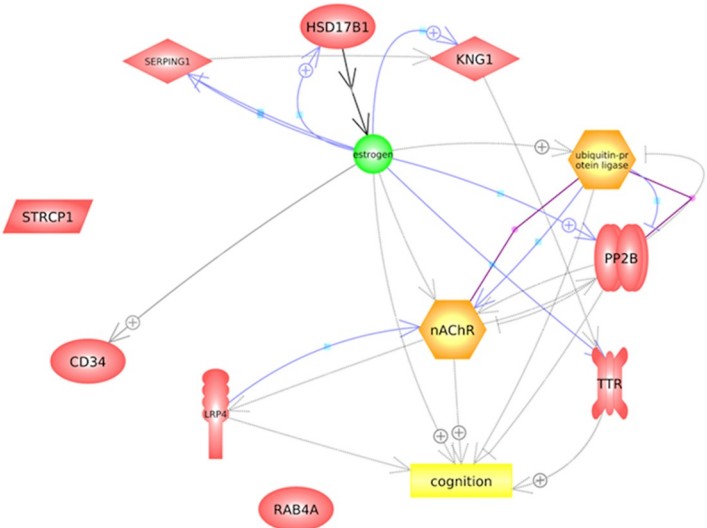

**Fig 4. Protein-protein interaction network produced by Pathway Studio.** The purple line indicates binding, the blue line indicates expression, the black line indicates chemical reactor, the solid gray indicates molecular transport, while the gray dotted line indicates regulation. The plus sign indicates positive regulation.

The WCST results in this study suggested that education may have an effect on attenuating cognitive sex differences. This is consistent with previous research demonstrating that sex differences in attention control to spatial cognition can be eliminated through short-term instruction or long-term experience in life [48–50]. Longer education led to increased cortical thickness in bilateral temporal, parietal, medial-frontal, sensory, and motor cortices in the adult brains [51]. Even after six weeks of multi-component cognitive control training, adolescents showed increased GMV in the right inferior frontal cortex (rlFC) [52]. In contrast, Rzezak et al. [53] discovered subtle or no differences in both whole-brain and regional GMV between participants with high and low education levels. Furthermore, a prior study found that females benefit more from certain training than males, and follow-up testing suggested that the gain was persistent [50]. This suggests a possible explanation for the effect of education on attenuating cognitive sex differences in WCST in terms of brain morphology: Formal education, which contains a curriculum for knowledge acquisition as well as physical and awareness development [54–56], promotes brain development in both sexes, particularly in the frontal cortex [57], which involved in attention and set-shifting in WCST [44, 58], and such effects may be accentuated in women, attenuating the cognitive sex differences in these two WCST scores. This idea was partially supported by earlier findings that education as well as closing the gap between males and females in educational opportunities, promote cognitive performance in females [18], and the skills required for efficient completion of the WCST may be closely associated with those taught and required in formal education [16]. In contrast, females did worse in the primary education group for both %Corrects and %Errors, with fewer correct responses and more errors. The total errors of WCST reflect non-specific cognitive impairments, and another research we conducted suggested that females might be more susceptible to the excitotoxicity induced by WCST [59]. This, combined with the loss of estrogen's neuroprotection effects after menopause [60–62], led to loss of brain environment homeostasis and impaired cognitive function in elderly female subjects [59].

On the other hand, healthy participants were thought to be more or less homogeneous on WCST performance because no sex difference was observed [63, 64]. Previous clinical research

has demonstrated that GMV is connected to WCST performance [65, 66]. Despite a male advantage in GMV, females have higher gray matter density than males throughout the brain, which may compensate for the lower GMV in females [67], leading to no significant sex differences in general intelligence [68].

Additionally, females outperformed males in score 1st Category in this study, but only in the higher education group. This is consistent with previous findings [16]. This score reflects initial conceptualization abilities, and numerous research findings support the idea that lateral PFC plays a central role in rule-based reasoning [69, 70], which further suggested that female brains, especially the PFC region, may be more sensitive to the boost effects of education. The other two WCST scores, Category Completed and PE, did not demonstrate significant sex differences in the current research, which contradicts earlier studies [16, 71]. Perhaps the relative low number of subjects following stratification by education in each education group prevents us from detecting sex differences in these two scores [16]. Interestingly, Cinciute et al. [17] identified significant sex-related neuronal activity in the PFC in healthy participants, but behavioral results of the WCST did not show sex bias. This contradiction arises because some functional brain maps of the WCST may be sensitive to neurophysiology heterogeneity which is not apparent or necessarily related to the behavior [17]. Although the standardization, validity and reliability of WCST as a stand-alone cognitive assessment in healthy adults of various ages and education levels have been established [72], a battery of cognitive tests is required to unveil the effects of education on different domains of cognition. Aside from education, other environmental factors that have been linked to cognitive sex differences include lifestyle factors [73], socio-economic status (SES) [74, 75], and cultural differences [76, 77]. Of these, SES has been proposed to be a factor in the discrepancy of sex difference in cognitive aging [75]. However, such factors were not investigated in the current study. To further understand the connections between environmental factors and cognitive sex differences, additional research with considering more factors related to changing sex discrepancy in cognitive function is required.

At the molecular level, the present study provided preliminary evidence which suggested that cholinergic signaling may mediate the effect of education on attenuating cognitive sex differences. Acetylcholine (ACh) is thought to be a neuromodulator in the central nervous system in adults, altering neuronal excitability, inducing synaptic plasticity, and coordinating the firing of groups of neurons [78]. For example, the firing of dopamine (DA) neurons located in the ventral tegmental area, as well as the release of DA by their projections in the striatal area, is tightly controlled by the cholinergic pathway mediated by nicotinic acetylcholine receptors (nAChRs) [79]. DA depletion has been linked to widespread impairment in connectivity between frontal cortex and striatum as well as worsened set-shifting ability during WCST [80], and this ability in WCST indicates the metric of learning, abstract reasoning, and problem solving; participants with better set-shifting have a better opportunity of achieving a higher correct rate [81]. Furthermore, earlier research found that experience alters the cholinergic pathway in the hippocampus of mice by interaction with a gene product and repeated learning specifically amplified the effect of such gene product expression on this cholinergic projection pathway [82]. Effects of cholinergic enhancement on experience-dependent plasticity have also been observed in healthy adult auditory cortex [83, 84], as well as their visual perceptual learning [85], and a subsequent follow-up study elucidated that pharmacological cholinergic enhancement on visual perceptual learning is long-lasting [86]. Baskerville and collaborators [87], on the other hand, investigated the effects of ACh input depletion from nucleus basalis on experience-dependent plasticity in the cortex of young adult male rats and discovered that cholinergic-depleted animals showed no significant plasticity response. This suggested that a cholinergic enhancement on learning-dependent plasticity is likely to be sex-specific, with females benefiting more.

Another finding of the present study is that nearly half of the DPEs interacted directly with estrogen. Not surprisingly, the cholinergic pathways are critical sites for estrogen in the brain [88], and multiple basic and preclinical research over the past decades has clearly demonstrated that basal forebrain cholinergic systems rely upon estradiol support for adequate functioning [89], and these cholinergic projections play an essential role in attentional processes and learning [90, 91]. Besides this, available evidence suggests many of the effects of estrogen on neuronal function and plasticity, as well as cognitive performance, are related to or dependent on interactions with such cholinergic projections [92–95]. Consistent with these findings, we detected an estrogen-nAChR3 interaction that is associated with cognitive performance; nAChR3 expression was upregulated in females and this upregulation was more apparent in the secondary and tertiary education groups.

Although the cholinergic-estrogen interaction offers evidence for the effect of education on attenuating cognitive sex differences, other inherent or external factors with similar effects cannot be ignored. Prior research indicated that in rats, the arginine vasopressin (AVP) deficiency caused by a mutation in the *Avp* gene eliminated sex differences in the extinction of conditioned taste aversion [96] and social reinforcement [97]. Except for education, one socio-cognitive factor was also proposed to abolish sex differences in mental rotation performance [98], the probable underlying mechanism might be confidence as a mediator by which sex stereotype exerts its influence. That is, when females are the target of sex stereotype, their working memory capacity decreased dramatically [99, 100], resulting in a drop in logical reasoning [101] and mathematical performance [102].

There are some limitations to our study. First, this study included subjects ranging from young adults to elderly, which could have increased the variability of the sample. Furthermore, the proportion of middle-aged and elderly subjects was much higher than that of young adults, with the young participants clustering in the higher education group. A further study based on young adults with a larger sample size is needed to replicate the current findings. Second, cognitive function and education are related, but not only because education influences cognition; higher intellectual function can lead to more education. Third, except for age, the other confounders such as socioeconomic status and occupation were not controlled for in this study, despite the fact that both potentially contribute to the sex difference in cognitive performance. Fourth, although a computerized version of the WCST was utilized in the study, the response time of the participants has not been examined, which could be a more sensitive measure to show the sex-related differences, and future research that measures both WCST sub-scores and response time is needed to better understand the cognitive sex differences in WCST. Fifth, in this study, neuronal acetylcholine receptor subunit alpha-3 (nAChR3) was detected in serum of the research subjects; however, due to the brain-blood barrier, the majority of the nAChRs may be synthesized locally; therefore, further research into the change of nAChRs expression levels in brain tissues induced by learning is required to better understand these results. Sixth, the cross-sectional design of this study makes drawing causal inferences about the relationships among the variables under investigation challenging. Longitudinal studies are needed in the future to track changes in cognitive performance and protein expression over time and to assess potential causal relationships. Last, owing to the moderate sample size, further generalizing the results of our study needs to be cautious.

## Conclusion

This study showed a preliminary connection between a cholinergic-estrogen interaction and the compensating effects of education on cognitive sex differences in a Thai healthy population. This suggested that learning experience is mediated by sex-dependent neural activity.

Education, for example, is associated with a differential modification of the cholinergic signaling in males and females, with women benefiting more. Once again, this implies that brain development is a result of genetic-environmental interaction, and future longitudinal research that takes into account more environmental factors such as lifestyle factors and medical conditions will be required to better understanding the environmental-biological mechanisms underlying cognitive sex performance.

## Supporting information

**S1 Table. Results of Wisconsin Card Sorting Test.**
(XLSX)

**S2 Table. Intensity of 11 differentially expressed proteins.**
(XLSX)

## Acknowledgments

The authors would like to thank all participants in this study. We would also like to thank the Faculty of Medical Science, Naresuan University and The National Center for Genetic Engineering and Biotechnology, Pathum Thani, Thailand for the facility supports.

## Author Contributions

**Conceptualization:** Chen Chen, Samur Thanoi, Sutisa Nudmamud-Thanoi.

**Data curation:** Chen Chen, Bupachad Khanthiyong, Sittiruk Roytrakul.

**Formal analysis:** Chen Chen, Sutisa Nudmamud-Thanoi.

**Methodology:** Chen Chen, Sawanya Charoenlappanit, Sittiruk Roytrakul.

**Project administration:** Sutisa Nudmamud-Thanoi.

**Supervision:** Gavin P. Reynolds, Samur Thanoi, Sutisa Nudmamud-Thanoi.

**Writing – original draft:** Chen Chen.

**Writing – review & editing:** Gavin P. Reynolds, Samur Thanoi, Sutisa Nudmamud-Thanoi.

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
