## [Decision Letter · Decision Letter 0]

10 Jan 2023

PONE-D-22-30636

Cholinergic-estrogen interaction underpins the effect of education on attenuating cognitive sex differences in a Thai healthy population

PLOS ONE

Dear Dr. Nudmamud-Thanoi,

Thank you for submitting your manuscript to PLOS ONE. After careful consideration, we feel that it has merit but does not fully meet PLOS ONE’s publication criteria as it currently stands. Therefore, we invite you to submit a revised version of the manuscript that addresses the points raised during the review process.

We look forward to receiving your revised manuscript.

Kind regards,

Vanessa Carels

Staff Editor

PLOS ONE

Journal Requirements:

   "CC was supported by the Naresuan Competitive Grants for International Students (NCG). We would like to thank Professor Gavin P. Reynolds, Biomolecular Sciences Research Centre, Sheffield Hallam University, UK, for his guidance and suggestions throughout the manuscript. We would also like to thank the facilities support from the Faculty of Medical Science, Naresuan University and The National Center for Genetic 

Engineering and Biotechnology, Pathum Thani, Thailand."

  "The authors received no specific funding for this work."

Additional Editor Comments:

The manuscript has been evaluated by two reviewers, and their comments are available below.

The reviewers have raised a number of concerns that need attention, and they request additional information on methodological aspects of the study and analyses. Further consideration will depend on your ability to thoroughly respond to all issues raised. 

Could you please revise the manuscript to carefully address the concerns raised?

Reviewers' comments:

Reviewer's Responses to Questions

**Comments to the Author**

1. Is the manuscript technically sound, and do the data support the conclusions?

Reviewer #1: Partly

Reviewer #2: Partly

2. Has the statistical analysis been performed appropriately and rigorously? 

Reviewer #1: No

Reviewer #2: Yes

3. Have the authors made all data underlying the findings in their manuscript fully available?

Reviewer #1: Yes

Reviewer #2: Yes

4. Is the manuscript presented in an intelligible fashion and written in standard English?

Reviewer #1: Yes

Reviewer #2: Yes

5. Review Comments to the Author

Reviewer #1: This study aims at examining the effects of education level on cognitive sex differences. The authors classify participants at different ages to three categories based on their education levels and then examine the differences in various WCST scores between males and females. They have also examined the sex-specific neurotransmission-related protein expression profiles and tried to link it to the sex-related differences in behavioural measurements. Understanding the differences in cognitive abilities between males and females and delineating the effects of environmental factors are important and therefore the main goal of this study has merit. However, there are various uncontrolled factors in this study, which make it inconclusive. The analytical approach also hinders reaching a clear conclusion.

Major points:

1- In this study, the education level (primary, secondary,…) has been considered as an isolated factor, however various other factors (such as socioeconomic status) might be strongly associated with the education level. The authors also cite previous studies indicating that socioeconomic status might significantly affect cognitive development and performance in cognitive tasks. This issue has not been controlled in this study.

2- The analytical approaches should be accompanied by direct comparison of the groups within the same statistical analyses (multi-factorial ANOVA). While assessing the evidence for a particular hypothesis for each separate group is informative, direct comparison of groups appears necessary. In other words, the effects of Age, Sex and Education factors should be examined with analytical approaches (same statistical analysis) to confirm the significance (or absence of significance) for the main effects and their interactions.

3- The examination of differentially expressed proteins between males and females, is informative, however it is simply a correlational analysis and does not establish any causal interaction between the cognitive sex difference and alterations in these proteins. Elsewhere in the manuscript a causative association is claimed “we discovered that cholinergic signalling may be at the root of the effect of education on attenuating cognitive sex differences” or “Cholinergic estrogen interaction underpins the effect of education on attenuating cognitive sex differences in a Thai healthy population”. However, the experimental approaches and results do not establish a causative link and conclusions are not supported.

Minor issues

1- Although they have used a computerized version of the WCST, the participants’ response time (RT) has not been measured/reported. RT could have been a more sensitive measure to show the sex-related differences (Feizpour et al. Psychology of Music 2018).

2- Literature review in the Introduction and Discussion needs to cite and discuss relevant studies in the context of the WCST.

3- Figure 1 can be improved in terms of clarity and quality. Currently, the data points cannot be easily detected.

Reviewer #2: This manuscript hypothesizes that education decreases the cognitive differences between genders. To examine this an experimental design was developed in humans, especially those who studied above primary level of education and those who didn’t. The authors use of Bayes factor as a statistic is impressive and the results that span behavioural, biological and bioinformatical domains showcases the strong experimental design. has

1. These findings could be discussed in light of following papers, as some studies have found marginal improvements with education and gender based changes in cognitive performances. - https://www.ncbi.nlm.nih.gov/pmc/articles/PMC4608774/ ; Efstathios D. Gennatas, Brian B. Avants, Daniel H. Wolf, Theodore D. Satterthwaite, Kosha Ruparel, Rastko Ciric, Hakon Hakonarson, Raquel E. Gur, Ruben C. Gur. Age-Related Effects and Sex Differences in Gray Matter Density, Volume, Mass, and Cortical Thickness from Childhood to Young Adulthood. The Journal of Neuroscience, 2017; 37 (20): 5065 DOI: 10.1523/JNEUROSCI.3550-16.2017

2. Lines 63-65 are not clear, What is the conclusion of that study?

3. Also, a population with age range of 22-70 was used here, this is a big range for age. As some are young adults and some are aging.

4. WISC is used in set shifting, capability of a person to change his attention based on new rules, it is not a memory test, Although it showcases frontal lobe function, especially the DLPFC, still many other forms of cognitive/memory testing could be done and would have elevated the paper. An explanation to this in the discussion would help, please cite other papers where WCST was used alone and the reliability of WCST for cognitive measurements as a stand-alone test.

5. Line 176-177, the age differences between groups, meaning the tertiary and secondary educated groups may just perform well because they are significantly lower in age compared to primary educated group?

6. The graphs are not consistent, In one graph, males are followed by females, while in others, it’s the reverse.

7. Also, did the authors make a comparison of females in tertiary education to primary education? Was there a difference? Careful examination of data with multiple comparisons are needed in primary vs. secondary educated females, males across all three parameters shown.

8. Also, a discussion on why the other two parameters of WCST were not showing results needs to be discussed? Please refer to any previous publications that showed such discrepancies if any? If not, please consider adding more numbers to the study.

6. PLOS authors have the option to publish the peer review history of their article (what does this mean?). If published, this will include your full peer review and any attached files.

Reviewer #1: No

Reviewer #2: **Yes: **Srinivasa P Kommajosyula

---

## [Author Response · Author response to Decision Letter 0]

9 Mar 2023

Respond to Editor comments:

Author response:

We have done the manuscript following PLOS ONE's style template.

Author response:

The details regarding participant consent have been added to the Methods and online submission information as “Written consent forms were obtained from the participants involved. The experimental protocols were approved by the Human Ethics Committee of Naresuan University (Naresuan University Institutional Review Board), COA No. 0262/2022”.

 "CC was supported by the Naresuan Competitive Grants for International Students (NCG). We would like to thank Professor Gavin P. Reynolds, Biomolecular Sciences Research Centre, Sheffield Hallam University, UK, for his guidance and suggestions throughout the manuscript. We would also like to thank the facilities support from the Faculty of Medical Science, Naresuan University and The National Center for Genetic 

Engineering and Biotechnology, Pathum Thani, Thailand."

 "The authors received no specific funding for this work."

Author response:

We have updated the funding statement and the details of funding have been included within the cover letter. The information on funding has been removed from the acknowledgment section. 

Author response:

We have provided the Data Availability statement in the cover letter as “All relevant data are within the manuscript and the supporting information files”.

Respond to reviewer comments:

5. Review Comments to the Author

Reviewer #1: This study aims at examining the effects of education level on cognitive sex differences. The authors classify participants at different ages to three categories based on their education levels and then examine the differences in various WCST scores between males and females. They have also examined the sex-specific neurotransmission-related protein expression profiles and tried to link it to the sex-related differences in behavioural measurements. Understanding the differences in cognitive abilities between males and females and delineating the effects of environmental factors are important and therefore the main goal of this study has merit. However, there are various uncontrolled factors in this study, which make it inconclusive. The analytical approach also hinders reaching a clear conclusion.

Major points:

1- In this study, the education level (primary, secondary,…) has been considered as an isolated factor, however various other factors (such as socioeconomic status) might be strongly associated with the education level. The authors also cite previous studies indicating that socioeconomic status might significantly affect cognitive development and performance in cognitive tasks. This issue has not been controlled in this study.

Author Response:

When comparing sex differences in WCST sub-scores at each educational level, age as a confounder was controlled. The model diagnostic was then performed to ensure that the model fits the data. Furthermore, the Bayes factor (BF) was employed as direct evidence to support the alternative hypothesis [ref 38]. This is done to ensure the validity of our findings. However, because other confounders such as socioeconomic status and profession were not controlled for, this study may have limitations, which have been noted in the revised manuscript. 

2- The analytical approaches should be accompanied by direct comparison of the groups within the same statistical analyses (multi-factorial ANOVA). While assessing the evidence for a particular hypothesis for each separate group is informative, direct comparison of groups appears necessary. In other words, the effects of Age, Sex and Education factors should be examined with analytical approaches (same statistical analysis) to confirm the significance (or absence of significance) for the main effects and their interactions.

Author response:

The results of multi-factorial ANOVA revealed that sex, age and education level (P≤0.05) had significant main effects on two of the five WCST-sub scores %Corrects and %Errors. Aside from that, there were significant interaction effects of sex by age (P＜0.01) on scores %Corrects, %Errors and Category Completed, as well as significant age by education interaction effect on 1st Category (P＜0.05). This information has been added to the revised manuscript (lines 195-201).

3- The examination of differentially expressed proteins between males and females, is informative, however it is simply a correlational analysis and does not establish any causal interaction between the cognitive sex difference and alterations in these proteins. Elsewhere in the manuscript a causative association is claimed “we discovered that cholinergic signalling may be at the root of the effect of education on attenuating cognitive sex differences” or “Cholinergic estrogen interaction underpins the effect of education on attenuating cognitive sex differences in a Thai healthy population”. However, the experimental approaches and results do not establish a causative link and conclusions are not supported.

Author response:

We agree that it is inappropriate to imply causality and have changed the manuscript throughout to avoid that suggestion. 

Minor issues

1- Although they have used a computerized version of the WCST, the participants’ response time (RT) has not been measured/reported. RT could have been a more sensitive measure to show the sex-related differences (Feizpour et al. Psychology of Music 2018).

Author response:

We appreciate the reviewer’s suggestion and agree this is a limitation which we have acknowledged in the discussion (lines 378-382). This will be used as a measure in further study.

2- Literature review in the Introduction and Discussion needs to cite and discuss relevant studies in the context of the WCST.

Author response:

The Introduction and Discussion of the manuscript have been revised, citing and discussing relevant WCST studies. 

3- Figure 1 can be improved in terms of clarity and quality. Currently, the data points cannot be easily detected.

 Author response:

The clarity and quality of the original figure should be no problem, with the data points can be easily seen. However, we have improved its clarity and updated it in the revised manuscript.

Reviewer #2: This manuscript hypothesizes that education decreases the cognitive differences between genders. To examine this an experimental design was developed in humans, especially those who studied above primary level of education and those who didn’t. The authors use of Bayes factor as a statistic is impressive and the results that span behavioural, biological and bioinformatical domains showcases the strong experimental design. Has

1. These findings could be discussed in light of following papers, as some studies have found marginal improvements with education and gender based changes in cognitive performances. - https://www.ncbi.nlm.nih.gov/pmc/articles/PMC4608774/ ; Efstathios D. Gennatas, Brian B. Avants, Daniel H. Wolf, Theodore D. Satterthwaite, Kosha Ruparel, Rastko Ciric, Hakon Hakonarson, Raquel E. Gur, Ruben C. Gur. Age-Related Effects and Sex Differences in Gray Matter Density, Volume, Mass, and Cortical Thickness from Childhood to Young Adulthood. The Journal of Neuroscience, 2017; 37 (20): 5065 DOI: 10.1523/JNEUROSCI.3550-16.2017

Author response:

We appreciate the reviewer’s suggestion, and these two papers have been studied and cited in the revised manuscript as ref.51 and ref.61. 

2. Lines 63-65 are not clear, What is the conclusion of that study?

Author response:

According to Bloomberg and others [ref 18], males in UK had higher fluency scores only in the low education group (less than high school diploma). There was evidence of greater fluency scores in females in the high education group (high school diploma and higher), particularly in those born later. This finding implied that schooling, as well as secular changes in education level across birth cohorts, had a role in determining cognitive performance in females. We have now included a clarification to this effect in the text (lines 71-76).

3. Also, a population with age range of 22-70 was used here, this is a big range for age. As some are young adults and some are aging.

Author response:

In an attempt to obtain a sample that is approximately representative of the population, the sample covers a range of subjects with varied ages, which could have increased the variability of the sample. To ensure the validity of the study, age was controlled in the process of the data analysis. This limitation has also been mentioned in the discussion of the revised manuscript (lines 369-374). 

4. WISC is used in set shifting, capability of a person to change his attention based on new rules, it is not a memory test, Although it showcases frontal lobe function, especially the DLPFC, still many other forms of cognitive/memory testing could be done and would have elevated the paper. An explanation to this in the discussion would help, please cite other papers where WCST was used alone and the reliability of WCST for cognitive measurements as a stand-alone test.

Author response:

The standardization, validity and reliability of WCST as a stand-alone cognitive assessment in healthy adults of various ages and education levels have been established [ref 66]. However, a battery of cognitive tests is required to unveil the effects of education on different domains of cognition. We have discussed this in the revised manuscript (lines 317-320).

5. Line 176-177, the age differences between groups, meaning the tertiary and secondary educated groups may just perform well because they are significantly lower in age compared to primary educated group?

Author response:

When comparing the WCST scores of subjects from the two educational groups. Females with a higher education level outperformed those in the primary education group in only two WCST sub-scores: %Corrects and %Errors. While there is limited evidence to show differences in WCST sub-scores in males across the two educational levels. Age as a covariate was controlled in the calculation. 

6. The graphs are not consistent, In one graph, males are followed by females, while in others, it’s the reverse.

Author response:

The graph in Figure 1 of this manuscript was created using the default setting of R package “flexplot”. We agree with the reviewer’s suggestion. So, the graphs have been revised.

7. Also, did the authors make a comparison of females in tertiary education to primary education? Was there a difference? Careful examination of data with multiple comparisons are needed in primary vs. secondary educated females, males across all three parameters shown.

Author response:

Yes, we did. Females with secondary and tertiary education performed significantly better in scores %Corrects and %Errors (BF＞100) than those in the lower education group, and moderate evidence for such difference in the score 1st Category (BF=4.55). While limited evidence indicated differences in scores of %Corrects (BF=4.79), %Errors (BF=4.43), and 1st Category (BF=3.48) in males across the two education levels. This has been added to the revised manuscript (lines 227-232).

8. Also, a discussion on why the other two parameters of WCST were not showing results needs to be discussed? Please refer to any previous publications that showed such discrepancies if any? If not, please consider adding more numbers to the study.

Author response:

The other two WCST scores, Category Completed and PE, did not demonstrate significant sex differences in the current research, which contradicts earlier studies [ref 16, 65]. Perhaps the relative low number of subjects following stratification by education in each education group prevent us from detecting sex differences in these two scores [ref 16] (lines 308-312). The limitation of the moderate sample size has been noted in the text (lines 387-388).

---

## [Decision Letter · Decision Letter 1]

17 Apr 2023

PONE-D-22-30636R1Cholinergic-estrogen interaction is associated with the effect of education on attenuating cognitive sex differences in a Thai healthy populationPLOS ONE

Dear Dr. Nudmamud-Thanoi,

Thank you for submitting your manuscript to PLOS ONE. After careful consideration, we feel that it has merit but does not fully meet PLOS ONE’s publication criteria as it currently stands. Therefore, we invite you to submit a revised version of the manuscript that addresses the points raised during the review process.

We look forward to receiving your revised manuscript.

Kind regards,

Thiago P. Fernandes, PhD

Academic Editor

PLOS ONE

Journal Requirements:

Additional Editor Comments:

Please respond to each comment AND highlight them.

Also enlist a native English speaker to edit your manuscript.

Reviewers' comments:

Reviewer's Responses to Questions

**Comments to the Author**

1. If the authors have adequately addressed your comments raised in a previous round of review and you feel that this manuscript is now acceptable for publication, you may indicate that here to bypass the “Comments to the Author” section, enter your conflict of interest statement in the “Confidential to Editor” section, and submit your "Accept" recommendation.

Reviewer #2: All comments have been addressed

Reviewer #3: All comments have been addressed

2. Is the manuscript technically sound, and do the data support the conclusions?

Reviewer #2: Yes

Reviewer #3: Yes

3. Has the statistical analysis been performed appropriately and rigorously? 

Reviewer #2: Yes

Reviewer #3: Yes

4. Have the authors made all data underlying the findings in their manuscript fully available?

Reviewer #2: Yes

Reviewer #3: Yes

5. Is the manuscript presented in an intelligible fashion and written in standard English?

Reviewer #2: Yes

Reviewer #3: Yes

6. Review Comments to the Author

Reviewer #2: All the raised queries have been addressed by the authors of this manuscript.

I thank the editor for the opportunity to review this study.

Reviewer #3: The article titled "Sex Differences in Cognitive Performance: Potential Compensatory Mechanisms Revealed by Proteomic Analysis" presents a study investigating sex differences in cognitive performance and potential compensatory mechanisms using proteomic analysis. The article is well-structured, providing detailed information on the study's background, methods, results, and discussion.

The study involved 135 subjects, 70 males, and 65 females, with a mean age of 57.75±10.35 years. The authors explored the effects of education on sex differences in cognitive performance by administering the Wisconsin Card Sorting Test (WCST) and comparing the results between males and females in different education groups. They found that education had potential compensatory effects on three out of the five WCST sub-scores: %Corrects, %Errors, and 1st Category. In the primary education group, males performed better in %Corrects and had lower %Errors, while females completed the first category in fewer trials. However, in the higher education group, sex differences in those scores reversed, with females performing better in %Corrects and males completing the first category in fewer trials. The authors suggest that education may play a role in compensating for sex differences in cognitive performance, although further research is needed to confirm this hypothesis.

In addition to the behavioral tests, the authors used a label-free quantitative proteomics approach to identify differentially expressed proteins (DEPs) between males and females. They found that 11 DEPs were present in both education groups, with all but one upregulated in females. The authors then investigated the protein-protein interactions of these DEPs and their relationship with cognitive function. They found that LRP4, TTR, TRIM38, and PP2B directly interacted with nAChR3, a subunit of nicotinic acetylcholine receptor that positively regulates cognition. On the other hand, CD34, SERPING1, HSD17B1, KNG1, and TTR were indirectly connected to the cholinergic pathways through interaction with estrogen, suggesting that the cholinergic-estrogen interaction may have an influence on cognitive processes.

The significance of this study lies in its exploration of the potential effects of education on sex differences in cognitive performance and their relationship with differentially expressed proteins between men and women. The study provides insights into the complex interactions between education, sex, and cognitive function, and sheds light on potential molecular mechanisms underlying sex differences in cognitive performance.

The findings of the study suggest that education may have compensatory effects on sex differences in cognitive performance, particularly in the WCST sub-scores of %Corrects, %Errors, and 1st Category. The study also identified 11 differentially expressed proteins between men and women, most of which were upregulated in females, and highlighted the potential roles of cholinergic-estrogen interactions in cognitive processes.

These results have important implications for understanding sex differences in cognitive function and developing interventions to improve cognitive performance in both sexes. By identifying the molecular mechanisms underlying these differences, future research may be able to develop targeted interventions that leverage the compensatory effects of education and modulate the cholinergic-estrogen pathway to improve cognitive function in both men and women.

Overall, the article provides valuable insights into the potential compensatory mechanisms underlying sex differences in cognitive performance. The use of proteomics analysis adds a novel dimension to the study, providing a more detailed understanding of the biological mechanisms underlying cognitive performance. However, the study has some limitations, including the relatively small sample size, the cross-sectional design, and the lack of information on other potential confounding factors such as lifestyle factors or medical conditions. Nevertheless, the study provides a valuable contribution to the literature on sex differences in cognitive performance and sets the stage for further research in this area.

Major concern

1. Sample size: The sample size of the study is relatively small, which limits the generalizability of the findings. Larger and more diverse samples would provide a more robust assessment of the relationships among cognitive performance, sex, education, and protein expression.

2. Methodological limitations: The study used a cross-sectional design, which makes it challenging to draw causal inferences about the relationships among the variables under investigation. Longitudinal studies would be useful for tracking changes in cognitive performance and protein expression over time and for evaluating potential causal relationships.

3. Statistical analysis: Although the study used Bayesian statistical methods, the presentation of the results could be improved by providing more detailed information on the Bayesian analyses, such as prior distributions and sensitivity analyses.

Minor concerns

1. Clearer headings and subheadings: The headings and subheadings in this article are a bit confusing and could be made clearer. For example, the section on demographic data could be titled "Demographic Characteristics of the Study Population" to better convey what the section is about.

2. Improved organization: The article would benefit from a clearer organization of ideas. For instance, the section on the effects of education on sex differences in cognitive performance could be restructured to present the findings in a more coherent manner. The section could start with an introduction explaining why the effects of education on cognitive performance were investigated, followed by a clear description of the results and a discussion of the findings.

3. Improved clarity: Some of the language used in the article is a bit difficult to understand. For example, the phrase "age differences between males and females in primary, secondary and tertiary education groups were supported by weak and substantial evidence, respectively" is confusing. A clearer way to say this might be "The study found that there were significant age differences between males and females in the secondary and tertiary education groups, but not in the primary education group."

4. More detailed discussion: The discussion section of the article is quite short and could benefit from more detailed analysis of the findings. For example, the article could explore why females in the higher education group performed better on certain cognitive tasks than males, whereas males in the lower education group performed better.

5. Address limitations: The limitations of the study should be addressed in the discussion section, and the authors should acknowledge any potential sources of bias or error in their study.

6. Clearer conclusions: The article's conclusions could be made clearer and more concise. The article could end with a summary of the main findings and their implications, along with suggestions for future research.

7. Consider alternative explanations: The authors should consider alternative explanations for their findings, and discuss other factors that may influence cognitive function, such as lifestyle factors, socio-economic status, and cultural differences.

Overall, the article contains valuable findings, but could benefit from some improvements in organization, clarity, and depth of analysis.

7. PLOS authors have the option to publish the peer review history of their article (what does this mean?). If published, this will include your full peer review and any attached files.

Reviewer #2: No

Reviewer #3: **Yes: **NAVEEN JAYAPRAKASH

---

## [Author Response · Author response to Decision Letter 1]

29 May 2023

Journal Requirements:

Regarding the Journal Requirements, we have reviewed all references and no references have been retracted.

Additional Editor Comments:

Please respond to each comment AND highlight them.

Also enlist a native English speaker to edit your manuscript.

Author response:

Regarding the editor's comments, we have responded all reviewer's comments seen below and the manuscript has been read and edited by the native English speaker.

6. Review Comments to the Author

Reviewer #3: The article titled "Sex Differences in Cognitive Performance: Potential Compensatory Mechanisms Revealed by Proteomic Analysis" presents a study investigating sex differences in cognitive performance and potential compensatory mechanisms using proteomic analysis. The article is well-structured, providing detailed information on the study's background, methods, results, and discussion.

The study involved 135 subjects, 70 males, and 65 females, with a mean age of 57.75±10.35 years. The authors explored the effects of education on sex differences in cognitive performance by administering the Wisconsin Card Sorting Test (WCST) and comparing the results between males and females in different education groups. They found that education had potential compensatory effects on three out of the five WCST sub-scores: %Corrects, %Errors, and 1st Category. In the primary education group, males performed better in %Corrects and had lower %Errors, while females completed the first category in fewer trials. However, in the higher education group, sex differences in those scores reversed, with females performing better in %Corrects and males completing the first category in fewer trials. The authors suggest that education may play a role in compensating for sex differences in cognitive performance, although further research is needed to confirm this hypothesis.

In addition to the behavioral tests, the authors used a label-free quantitative proteomics approach to identify differentially expressed proteins (DEPs) between males and females. They found that 11 DEPs were present in both education groups, with all but one upregulated in females. The authors then investigated the protein-protein interactions of these DEPs and their relationship with cognitive function. They found that LRP4, TTR, TRIM38, and PP2B directly interacted with nAChR3, a subunit of nicotinic acetylcholine receptor that positively regulates cognition. On the other hand, CD34, SERPING1, HSD17B1, KNG1, and TTR were indirectly connected to the cholinergic pathways through interaction with estrogen, suggesting that the cholinergic-estrogen interaction may have an influence on cognitive processes.

The significance of this study lies in its exploration of the potential effects of education on sex differences in cognitive performance and their relationship with differentially expressed proteins between men and women. The study provides insights into the complex interactions between education, sex, and cognitive function, and sheds light on potential molecular mechanisms underlying sex differences in cognitive performance.

The findings of the study suggest that education may have compensatory effects on sex differences in cognitive performance, particularly in the WCST sub-scores of %Corrects, %Errors, and 1st Category. The study also identified 11 differentially expressed proteins between men and women, most of which were upregulated in females, and highlighted the potential roles of cholinergic-estrogen interactions in cognitive processes.

These results have important implications for understanding sex differences in cognitive function and developing interventions to improve cognitive performance in both sexes. By identifying the molecular mechanisms underlying these differences, future research may be able to develop targeted interventions that leverage the compensatory effects of education and modulate the cholinergic-estrogen pathway to improve cognitive function in both men and women.

Overall, the article provides valuable insights into the potential compensatory mechanisms underlying sex differences in cognitive performance. The use of proteomics analysis adds a novel dimension to the study, providing a more detailed understanding of the biological mechanisms underlying cognitive performance. However, the study has some limitations, including the relatively small sample size, the cross-sectional design, and the lack of information on other potential confounding factors such as lifestyle factors or medical conditions. Nevertheless, the study provides a valuable contribution to the literature on sex differences in cognitive performance and sets the stage for further research in this area.

Major concern

1. Sample size: The sample size of the study is relatively small, which limits the generalizability of the findings. Larger and more diverse samples would provide a more robust assessment of the relationships among cognitive performance, sex, education, and protein expression.

Author response: 

We discussed the limitations of this study in the last paragraph of the Discussion section, which included the moderate sample size (line 409), and the proportion of middle-aged and elderly subjects was much higher than that of young adults, with the young participants clustering in the higher education group (lines 389-391). We agree that a larger and more diverse sample size would provide more robust results, and had revised our manuscript to show that our findings provided a preliminary indication of potential compensatory effects of education underlying cognitive sex difference attenuation (lines 41-43), and that further generalizing the results of our study should be done with caution (lines 409-410).

2. Methodological limitations: The study used a cross-sectional design, which makes it challenging to draw causal inferences about the relationships among the variables under investigation. Longitudinal studies would be useful for tracking changes in cognitive performance and protein expression over time and for evaluating potential causal relationships.

Author response: 

We appreciate the reviewer’s suggestion and agree this is a limitation which we have acknowledged in the discussion (lines 405-409). This will be used as the methodology in further study.

3. Statistical analysis: Although the study used Bayesian statistical methods, the presentation of the results could be improved by providing more detailed information on the Bayesian analyses, such as prior distributions and sensitivity analyses.

Author response: 

The default prior used in the general linear model is the Jeffreys-Zellner-Siow (JZS) priors [ref 40], with a default r value of 0.353 applied. After sensitivity analyses [ref 39], the results remain robust. This information has been added to the revised manuscript (lines 208-209, 221-222).

Minor concerns

1. Clearer headings and subheadings: The headings and subheadings in this article are a bit confusing and could be made clearer. For example, the section on demographic data could be titled "Demographic Characteristics of the Study Population" to better convey what the section is about.

Author response: 

We appreciate the reviewer’s suggestion and changed this subheading accordingly (line 184). 

2. Improved organization: The article would benefit from a clearer organization of ideas. For instance, the section on the effects of education on sex differences in cognitive performance could be restructured to present the findings in a more coherent manner. The section could start with an introduction explaining why the effects of education on cognitive performance were investigated, followed by a clear description of the results and a discussion of the findings.

Author response: 

The rationale of why the effects of education on cognitive performance has been presented on the Introduction section (lines 70-77); the sub-section “Effects of education on sex differences in cognitive performance” in the Results section described the findings from our data analysis (line 208-236), and the related discussion was made in the Discussion section (lines 266-331).

3. Improved clarity: Some of the language used in the article is a bit difficult to understand. For example, the phrase "age differences between males and females in primary, secondary and tertiary education groups were supported by weak and substantial evidence, respectively" is confusing. A clearer way to say this might be "The study found that there were significant age differences between males and females in the secondary and tertiary education groups, but not in the primary education group."

Author response: 

We appreciate the reviewer’s suggestion and revised this unclear sentence accordingly (lines 187-189).

4. More detailed discussion: The discussion section of the article is quite short and could benefit from more detailed analysis of the findings. For example, the article could explore why females in the higher education group performed better on certain cognitive tasks than males, whereas males in the lower education group performed better.

Author response: 

In the Discussion section, why females in the higher education group performed better than males on certain cognitive tasks than males has been discussed (lines 287-299, 315-319). Regarding to why males in the lower education group performed better, the total errors of WCST reflect non-specific cognitive impairments, and another study we conducted suggested that females might be more susceptible to the excitotoxicity induced by WCS [ref 59]. This, combined with the loss of estrogen’s neuroprotection effects after menopause [ref 60, 61, 62], led to loss of brain environment homeostasis and impaired cognitive function in elderly female subjects [ref 59], as indicated by their cognitive performance (more errors). This has been added in the Discussion section (lines 299-306).

5. Address limitations: The limitations of the study should be addressed in the discussion section, and the authors should acknowledge any potential sources of bias or error in their study.

Author response: 

We have stated the limitations of this research in the last paragraph of the Discussion section and acknowledged potential sources of bias or error in the study (lines 387-410).

6. Clearer conclusions: The article's conclusions could be made clearer and more concise. The article could end with a summary of the main findings and their implications, along with suggestions for future research.

Author response: 

We appreciate the reviewer’s suggestion and improved the conclusion of this study (lines 413-415, 419-422).

7. Consider alternative explanations: The authors should consider alternative explanations for their findings, and discuss other factors that may influence cognitive function, such as lifestyle factors, socio-economic status, and cultural differences.

Author response: 

We appreciate the reviewer’s suggestion. Aside from education, other environmental factors that have been linked to cognitive sex differences include lifestyle factors [ref 73]，socio-economic status (SES) [ref 74, 75], and cultural differences [ref 76, 77]. Of these, SES has been proposed to be a factor in the discrepancy of sex difference in cognitive aging [ref 75]. However, such factors were not investigated in the current study. To further understand the connections between environmental factors and cognitive sex differences, additional research considering more factors related to the sex discrepancy in cognitive function is required. This has been added in the revised manuscript (lines 331-338).

Overall, the article contains valuable findings, but could benefit from some improvements in organization, clarity, and depth of analysis.

---

## [Editor Report · Decision Letter 2]

31 May 2023

Cholinergic-estrogen interaction is associated with the effect of education on attenuating cognitive sex differences in a Thai healthy population

PONE-D-22-30636R2

Dear Dr. Nudmamud-Thanoi,

We’re pleased to inform you that your manuscript has been judged scientifically suitable for publication and will be formally accepted for publication once it meets all outstanding technical requirements.

Kind regards,

Thiago P. Fernandes, PhD

Academic Editor

PLOS ONE

Additional Editor Comments (optional):

By my own reading, the authors addressed all the raised concerns.
---

## [Editor Report · Acceptance letter]

11 Jul 2023

PONE-D-22-30636R2 

Cholinergic-estrogen interaction is associated with the effect of education on attenuating cognitive sex differences in a Thai healthy population 

Dear Dr. Nudmamud-Thanoi:

I'm pleased to inform you that your manuscript has been deemed suitable for publication in PLOS ONE. Congratulations! Your manuscript is now with our production department. 

Kind regards, 

on behalf of

Dr. Thiago P. Fernandes 

Academic Editor

PLOS ONE